# OpenReview forum: "RECAP: Reproducing Copyrighted Data from LLMs Training with an Agentic Pipeline"
_ICLR.cc/2026/Conference — Submitted to ICLR 2026_

### Official Review · Reviewer_pcch · 2025-10-29

**Soundness:** 2
**Presentation:** 3
**Contribution:** 2
**Rating:** 4
**Confidence:** 4

**Summary:**

This paper introduces RECAP, an agentic framework designed to extract memorized training data from Large Language Models (LLMs), addressing concerns about the illegal distribution of copyrighted material. The core contribution lies in its Jailbreaker component, which circumvents model alignment, and a Feedback Agent that creates an iterative extraction loop. The method is evaluated on a new benchmark, EchoTrace, comprising copyrighted books and scientific papers, and is accompanied by a detailed ablation study. While the paper tackles a significant and timely problem, the contributions feel incremental. The motivation for a new benchmark is unclear, and the experimental results are not fully compelling due to the choice of closed-source models and a lack of comparison on established benchmarks.

**Strengths:**

Originality: The primary originality lies in the iterative, agent-based design, specifically the Feedback Agent that refines extraction attempts based on previous failures. The Jailbreaker component, while sharing similarities with existing divergence attacks, is leading to the performance improvement.

Quality: The paper includes a thorough ablation study that analyzes the effectiveness and efficiency of RECAP's components, which is a valuable contribution for practitioners and future research.

Clarity: The paper is generally well-written and the RECAP framework is described clearly.

Significance: The problem of copyright infringement via LLM memorization is a critical issue for the AI community. Providing tools to audit and measure this risk is of high significance.

**Weaknesses:**

Novelty of Core Component: The application of the Jailbreaker, a core novelty, appears similar to divergent attacks proposed in prior work (e.g., Nasr et al., 2023), which also force models to deviate from their alignment. The paper would be strengthened by a clearer distinction and discussion of how this component differs from or builds upon existing attack paradigms.

Benchmark Justification and Scope: The creation of the EchoTrace benchmark is not sufficiently motivated. The field has well-established benchmarks for training data extraction (e.g., the Model Extraction Benchmark, the-stack-smol used by Wang et al., 2024). The choice to create a new one requires justification, especially since its composition (only 35 books and 20 papers, with short 40-token target sequences) may limit the generalizability of the findings. An ablation on the length of the extractable data is notably absent.

Experimental Setup on Closed-Source Models: The evaluation relies heavily on closed-source models (e.g., GPT-4.1, Gemini-2.5-Pro, Claude-3.7) for which the exact training data composition is unknown. This introduces a fundamental validity issue, as it is impossible to confirm whether the EchoTrace sources were actually in the model's training set. The choice of these models over open-weight alternatives is not adequately justified and weakens the evidence for RECAP's efficacy.

Comparative Evaluation: The results are less compelling due to the lack of comparison on established benchmarks used by contemporary work. For instance, comparing against Dynamic Soft Prompting (Wang et al., 2024) on the same test sets (e.g., the-stack-smol) would provide a more direct and convincing performance assessment.

**Questions:**

1. What was the specific motivation for creating the new EchoTrace benchmark instead of using or extending well-established datasets like the Pile, the Stack, or its derivatives? Were there specific limitations in these existing benchmarks that EchoTrace aims to address?

2. Could the authors extend their experimental results to include the test sets used by Wang et al., 2024 for Dynamic Soft Prompting? This would allow for a more direct and fair comparison with a closely related state-of-the-art method.

3. The paper mainly focuses on closed-source models. What was the reasoning behind this choice? Could the authors also evaluate RECAP on open-source models with known training data (e.g., GPT-Neo, Pythia) to conclusively verify that the extracted text was indeed part of the training corpus? This would significantly strengthen the validity of the claims.

4. There is a wrong citation of RedPajama (194).

---

> ### Author Response · Authors · 2025-11-20
> **Official Comment by Authors (Part 1)**
>
> Dear Reviewer,
>
> We sincerely appreciate the time and effort dedicated to our work. Below, we outline our responses to the comments and questions you have presented. We would also like to note that we have already updated the manuscript to incorporate many of your suggestions.
>
> > *Novelty of Core Component: The application of the Jailbreaker, a core novelty, appears similar to divergent attacks proposed in prior work (e.g., Nasr et al., 2023), which also force models to deviate from their alignment. The paper would be strengthened by a clearer distinction and discussion of how this component differs from or builds upon existing attack paradigms.*
> >
>
> Nasr’s approach was effective at making a model more prone to accept prompts, but the overall approach provided limited control over what the model ultimately outputted, which was reasonable given their broader data-scale objectives. In contrast, our Jailbreaker is designed to *steer* the model toward completing a *specific* crafted prompt, enabling controlled and targeted extraction rather than simply waiting for spontaneous verbatim leakage. This distinction is crucial because our goal is reliable reproduction of a chosen passage, not general divergence from alignment.
>
> > *Benchmark Justification and Scope: The creation of the EchoTrace benchmark is not sufficiently motivated.* What was the specific motivation for creating EchoTrace instead of using or extending well-established datasets like the Pile, the Stack, or its derivatives?  *The choice to create a new one requires justification, especially since its composition (only 35 books and 20 papers, with short 40-token target sequences) may limit the generalizability of the findings. An ablation on the length of the extractable data is notably absent.*
> >
>
> We recognize that several benchmarks already exist for evaluating targeted data extraction. However, **EchoTrace is complementary rather than redundant**, as it enables a different and important form of memorization analysis. We outline the key distinctions below:
>
> **(A) EchoTrace measures in-book, domain-coherent memorization rather than isolated heterogeneous snippets**
>
> Benchmarks such as the Model Extraction Benchmark (subset of the Pile) evaluate targeted extraction on *highly heterogeneous* data where each sample is often an independent, stand-alone sentence. This design is well-suited for probing whether models memorize arbitrary short snippets (e.g., “My phone number is …”). However, it is not designed to capture **how memorization manifests within coherent long-form sources** such as books or research papers. EchoTrace specifically targets this setting by clustering extraction targets within full works, enabling analysis of memorization patterns at the document level.
>
> **(B) Clarification on passage length and evaluation spans**
> Another key distinction from benchmarks like the Model Extraction Benchmark is that EchoTrace does **not** expect models to reproduce only x-token sequences. Our true targets are substantially long passages, often spanning hundreds of tokens. In our case, the 40-token spans are used *solely* as standardized, comparable evaluation windows. A correct long-form reconstruction naturally yields multiple matching 40-token spans. We have included a clearer analysis of the length and structure of these extractable passages in the revised version.
>
> > *Could the authors extend their experimental results to include the test sets used by Wang et al., 2024?* (Model Extraction Benchmark and the-stack-smol)
> >
>
> We appreciate the suggestion. However, we believe the benchmarks used by Wang et al. differ in key ways from the setting that RECAP is designed for: (a) The Model Extraction Benchmark consists of very short and highly heterogeneous snippets, where applying RECAP’s summarization and feedback agents to only a few sentences would likely introduce too much paraphrasing or other unintended contamination. (b) The-stack-smol is a code-generation benchmark, where valid continuations are far less open-ended than in natural language and may be driven more by syntactic constraints than by genuine document-level recall. Additionally, both benchmarks evaluate models using ~50-token completions, whereas RECAP is intended for long-form, domain-coherent extraction. For these reasons, although running RECAP on these benchmarks is certainly possible, we believe it would not produce results that are directly comparable or particularly informative for the long-form memorization setting that EchoTrace aims to study.

---

> > ### Author Response · Authors · 2025-11-20
> > **Official Comment by Authors (Part 2)**
> >
> > > Experimental Setup on Closed-Source Models: The evaluation relies heavily on closed-source models (e.g., GPT-4.1, Gemini-2.5-Pro, Claude-3.7) for which the exact training data composition is unknown. This introduces a fundamental validity issue, as it is impossible to confirm whether the EchoTrace sources were actually in the model's training set. The choice of these models over open-weight alternatives is not adequately justified and weakens the evidence for RECAP's efficacy. The paper mainly focuses on closed-source models. What was the reasoning behind this choice?
> > >
> >
> > We do rely more on closed-source models, but we also evaluate RECAP on modern open-weight models (DeepSeek-V3 and Qwen3-32B), and we do observe the same separation between public-domain/copyrighted books and non-training books there as in the closed-source ones. We always see that non-training titles consistently show near-zero extraction, which reinforces that RECAP is surfacing genuine memorization rather than introducing contamination. Our focus on closed-source models is mostly practical: these are the systems actually deployed at scale, where copyright compliance questions matter most, and they are also the models whose capabilities make long-form extractable memorization observable, a point we expand on in the next point.
> >
> > > Could the authors also evaluate RECAP on open-source models with known training data (e.g., GPT-Neo, Pythia) to conclusively verify that the extracted text was indeed part of the training corpus? This would significantly strengthen the validity of the claims.
> > >
> >
> > We agree that evaluating RECAP on models with known training data is valuable. We believe that GPT-Neo and Pythia are released only as base models, and up to our knowledge do not have official instruction-tuned variants. Since RECAP relies on the model’s ability to follow multi-step guidance, running it on non-instruct models would make the results difficult to interpret as poor extraction quality would likely come from limited instruction-following rather than from the absence of the target text in the training data.
> >
> > With that said, following the spirit of the suggestion, we instead evaluated RECAP on **OLMo-2-32B**, a fully open model with transparent training data and an instruct-tuned variant (Appendix Q). We are currently running RECAP across the entire EchoTrace benchmark, for which the (i) public-domain are all in the training data (the authors explicitly mention to train on Project Gutenberg), (ii) copyrighted bestsellers not publicly listed in the training-data sources but widely redistributed online, and (iii) post-cutoff non-training books.
> >
> > At the moment OLMo-2-32B exhibits the **same pattern** observed in our main results: strong extraction for known training data, near-zero extraction for guaranteed unseen books, and “training-like” extraction for illegally redistributed copyrighted titles. These results show that **RECAP reliably separates true exposures from non-training data** on a model with transparent training sources, and that OLMo-2-32B’s behavior aligns with the patterns observed across all other models evaluated.
> >
> > We are in the process of completing the full experiment and everything is proceeding as expected, but we need a few more days for all runs to finish. The updated manuscript already includes the **preliminary** OLMo-2-32B results in Appendix Q, and we will incorporate the complete results as soon as the evaluation concludes.
> >
> > > There is a wrong citation of RedPajama (194).
> > >
> >
> > Thank you for pointing this out, we fixed it.
> >
> > ---
> >
> > **Conclusions**
> >
> > We hope that our answers have addressed your concerns, and thank you once again for your valuable feedback.
> >
> > Please let us know if any further clarification or additional information is needed from our end.

---

> > > ### Author Response · Authors · 2025-11-23
> > > **Official Comment by Authors (Part 3)**
> > >
> > > Dear Reviewer,
> > >
> > > Following up on our previous comment, we just wanted to inform you that the OLMo-2-32B evaluation has now fully completed, and Appendix Q has been updated with the final results which remained consistent with the preliminary findings.
> > >
> > > We thank you once again for the insightful feedback, and we are happy to provide any further clarification if needed.

---

### Official Review · Reviewer_os15 · 2025-10-30

**Soundness:** 3
**Presentation:** 3
**Contribution:** 3
**Rating:** 6
**Confidence:** 5

**Summary:**

This paper addresses the critical challenge of verifying memorized training data in Large Language Models (LLMs) when training data inspection is unavailable. It proposes RECAP, an agentic pipeline featuring a feedback-driven iterative loop and a jailbreaking module, designed to elicit and verify verbatim memorized content from LLMs.Additionally, the paper introduces EchoTrace, a novel benchmark encompassing 35 full-length books and 20 arXiv research papers, totaling over 70,000 40-token passages for evaluation. Experimental results demonstrate RECAP's superiority: it achieves an average ROUGE-L score of 0.46 for copyrighted content across four model families, improves GPT-4.1’s copyrighted text extraction ROUGE-L from 0.38 to 0.47, and ensures no significant contamination from non-training data. The work also includes cost optimization and ethical considerations to avoid copyright misuse.

**Strengths:**

1.The feedback loop (via Feedback Agent) iteratively refines extractions without injecting excessive external information, reducing false positives.
2.The jailbreaking module effectively circumvents alignment-induced refusals, addressing a key limitation of prior methods like Prefix-Probing and Dynamic Soft Prompting (DSP).The hybrid memorization score filtering balances extraction quality and cost efficiency, addressing the practicality of iterative pipelines.
3.The experiments are comprehensive and well-designed:Cover multiple models and domains, ensuring generalizability. Include ablation studies and cost analysis, providing actionable insights for real-world use.

**Weaknesses:**

1.The benchmark overrepresents popular works for both public domain and copyrighted categories. This may overestimate RECAP's performance on less mainstream, rarely scraped texts—critical for assessing real-world applicability.
2.Non-training data is limited to 5 books released in 2025, with no diversity in genre or timeframes. This makes it hard to validate RECAP's robustness against false positives across varied non-training scenarios.
3.The jailbreaking module relies on a single static hand-crafted prompt, which the authors acknowledge may fail as LLMs’alignment updates advance. No comparison with dynamic jailbreaking methods is provided, leaving unclear whether static prompts are optimal or just a convenient choice.
4.The module’s effectiveness is only measured by refusal rates, not by whether jailbroken outputs introduce noise or semantic distortions, which could undermine extraction reliability.

**Questions:**

1.EchoTrace’s focus on popular works may overestimate RECAP's performance. Do you plan to expand the benchmark to include non-mainstream texts and validate RECAP's effectiveness on these?
2. The jailbreaking module uses a static prompt. Have you tested dynamic jailbreaking methods and compared their success rate, extraction quality, and robustness to future LLM alignment updates?
3.RECAP is not evaluated on open-source LLMs. Do you expect similar performance on these models, or would the pipeline require modifications ?
4.The paper omits comparisons with 2024–2025 state-of-the-art methods . Do you plan to replenish these comparisons, and if so, what preliminary insights can you share about RECAP's relative performance?

---

> ### Author Response · Authors · 2025-11-20
> **Official Comment by Authors (Part 1)**
>
> Dear Reviewer,
>
> We appreciate the time and effort invested in reviewing our paper. Below, we clarify your comments. We would also like to note that we have already updated the manuscript to incorporate many of your suggestions.
>
> > *The benchmark over-represents popular works for both public domain and copyrighted categories. This may overestimate RECAP’s performance on rarely scraped texts, critical for assessing real-world applicability. Do you plan to expand the benchmark to include non-mainstream texts?*
> >
>
> We agree with this observation, and we explicitly acknowledge this limitation in Appendix A. We emphasize that a strong RECAP score provides strong evidence of memorization, while a weak RECAP score **does not support the inverse conclusion** (i.e., that the text was never in training). We believe this asymmetric interpretability is still very desirable in practice, since RECAP does not seem to introduce false *membership* signals.
>
> Regarding real-world applicability: the most concerning copyright issues arise for *popular, widely scraped* works, because those are typically memorized at a scale that produces verbatim or near-verbatim leakage. Rarely scraped texts are significantly less prone to memorization, which naturally reduces the risk of verbatim extraction.
>
> That said, we agree that it is valuable to understand RECAP’s behavior on genuinely low-exposure, non-mainstream works. **We added an ablation experiment covering this scenario in Appendix R.**
>
> > *Non-training data is limited to 5 books released in 2025, with little genre or temporal diversity. This limits the assessment of RECAP’s robustness against false positives.*
> >
>
> For temporal diversity, our choice was constrained by the need for *guaranteed* non-training data when evaluating closed-source models. Books published after the model’s cutoff date are the only way to ensure this with certainty.
>
> Regarding content diversity: we intentionally kept the non-training books stylistically close to the rest of the benchmark. Using genres that differ strongly in structure or tone (e.g., textbooks or poetry) could change the model’s behavior for reasons unrelated to training exposure. By keeping genres aligned, we believe that we can more clearly assess whether the model treats genuinely unseen books differently.
>
> As for the number of books: we stopped at 5 because, as expected, none of them yielded meaningful extractable passages. Once it became clear that all models consistently failed to reproduce any content, expanding the set further provided little additional signal.
>
> > *The jailbreaking module relies on a single static, hand-crafted prompt. No comparison with dynamic jailbreakers is provided. Have you tested dynamic methods, and how do they compare with respect to success rate, extraction quality, and robustness to future updates?*
> >
>
> It is correct that our jailbreaking module uses one static prompt. We did not evaluate fully dynamic prompting approaches since the static prompt already yields near-maximum success rates across all models we tested, suggesting that the headroom for improvement via dynamic methods is small. Our code implementation already includes a second jailbreak prompt, and users can easily add more jailbreaks and extend this component it into a into dynamic module if desired.
>
> For these reasons, we agree a dynamic module would be valuable for robustness against future alignment updates but we do not expect it to materially affect the findings we reported in this paper.
>
> > *The module’s effectiveness is only measured by refusal rates, not by whether jailbroken outputs introduce noise or semantic distortions, which could undermine extraction reliability.*
> >
>
> We believe we partially address this point already. In Section U.1, we present a controlled test where we remove the Verbatim Verifier, forcing the Jailbreaking module to activate on every iteration. This stress-tests the jailbreak prompt’s influence on extraction quality. The results show a **slight degradation in performance**, indicating that the jailbreak prompt, while effective at reducing refusals, can introduce additional complexity when overused.
>
> We acknowledge that the current section may underplay this finding and we updated its framing to make this analysis more explicit.

---

> > ### Author Response · Authors · 2025-11-20
> > **Official Comment by Authors (Part 2)**
> >
> > > *RECAP is not evaluated on open-source LLMs. Do you expect similar performance on these models, or would the pipeline require modifications ?*
> > >
> >
> > We do evaluate RECAP on open-source models: specifically DeepSeek and Qwen-32B. Our findings suggest that the dominant factor for extractability is *model size*, not whether the model is open- or closed-source.
> >
> > The pipeline itself does not require any substantial modification for different open models. Any model that can be hosted through vLLM should be integrated in a plug-and-play manner.
> >
> > In fact, we added an entire new model experiments with a fully Open-Source model (OLMo-32B) which we also placed in Appendix Q.
> >
> > > *The paper omits comparisons with 2024–2025 state-of-the-art methods . Do you plan to replenish these comparisons, and if so, what preliminary insights can you share about RECAP's relative performance?*
> > >
> >
> > To our knowledge, related techniques from 2024–2025 (e.g., Min-K-Prob++ [1]) focus on *training data detection* rather than *training data extraction*. Since these address different problem settings, placing them side by side with RECAP would risk a misleading comparison. We are not aware of extractor-focused methods newer than DSP; if you have particular approaches in mind, we would gladly review them and include a discussion or baseline when feasible.
> >
> > [1] Zhang, J., Sun, J., Yeats, E., Ouyang, Y., Kuo, M., Zhang, J., ... & Li, H. (2024). Min-k%++: Improved baseline for detecting pre-training data from large language models.
> >
> > ---
> >
> > **Conclusions**
> >
> > We hope that our responses clarify the concerns raised and demonstrate the validity and value of our work. We thank you for the insightful feedback and we are happy to further clarify should you have any other questions.

---

### Official Review · Reviewer_Ytx5 · 2025-11-01

**Soundness:** 4
**Presentation:** 4
**Contribution:** 3
**Rating:** 8
**Confidence:** 4

**Summary:**

1. The paper tackles the task of extracting memorized data from LLM pretraining corpora to provide verifiable evidence of what models have seen during training.
2. It addresses two key challenges: (i) modern alignment safeguards that cause models to refuse reproducing even public-domain content, and (ii) the limited recall of single-iteration prompting methods that fail to elicit complete memorized passages.
3. The proposed agentic RECAP pipeline, achieves the highest ROUGE-L scores across all tested models and datasets.

**Strengths:**

1. The EchoTrace dataset is a valuable resource for future work. It covers diverse text types (public-domain, copyrighted, and unseen books). The segmentation and event summaries make it easy to test new elicitation or membership-inference methods.
2. The proposed RECAP method directly addresses both identified challenges. The results are clear and statistically grounded, showing strong and consistent improvements across four major model families.

**Weaknesses:**

In Prefix-Prompting baselines, longer or more detailed prefixes can sometimes lead to stronger verbatim reproduction. It would strengthen the paper if the authors analyzed whether prompt length differences contribute to RECAP’s performance gains.

**Questions:**

Could RECAP framework regulate prompt length within its agentic loop?

---

> ### Author Response · Authors · 2025-11-20
>
> Dear Reviewer,
>
> We greatly appreciate the time and effort you invested in reviewing our paper. We also appreciate the paper suggestion. Below, we provide clarification to your comments.
>
> > In Prefix-Prompting baselines, longer or more detailed prefixes can sometimes lead to stronger verbatim reproduction. It would strengthen the paper if the authors analyzed whether prompt length differences contribute to RECAP’s performance gains.
> >
>
> We appreciate this suggestion. We do believe, however, that we already addressed this question indirectly in Appendix V. In particular, Figure 20(a) presents an ablation on the number of tokens used by each method, which effectively reflects the prompt length of each approach. There, we observe that methods that consume more tokens also have higher extraction performance, which intuitively makes sense given that they can provide better contextual grounding for the models.
>
> > Could RECAP framework regulate prompt length within its agentic loop?
> >
>
> Yes, we believe that prompt length can be explicitly controlled, and the natural point of intervention is the Section Summary Agent. This module determines how much contextual information is included in the dynamic soft prompt that accompanies each extraction attempt. Shorter summaries make the prompt closer in spirit to Prefix-Probing, while longer summaries enrich the semantic guidance used by the extraction model.
>
> That said, we are aware that providing too much context increases the risk of inadvertently injecting information that could bias the extraction. For this reason, RECAP uses bullet-style summaries rather than high-granularity descriptions.
>
> ---
>
> **Conclusions**
>
> We hope that our answers have addressed your concerns, and thank you once again for your valuable feedback. Please let us know if any further clarification or additional information is needed from our end.

---

### Official Review · Reviewer_p8p2 · 2025-11-01

**Soundness:** 3
**Presentation:** 2
**Contribution:** 3
**Rating:** 4
**Confidence:** 3

**Summary:**

The work proposes a method called RECAP to test or verify whether a large language model (LLM) has memorized specific text data during training—particularly potentially copyrighted material, such as full books. To enable this verification, we construct a new dataset comprising public domain works, copyrighted books, and non-training new books, categorized by their likelihood of being memorized, and further integrate research papers into the evaluation.

**Strengths:**

- The dataset used is comprehensive enough to effectively demonstrate that the proposed method can successfully guide the LLM to reveal memorized content.

- The experiments are thorough and examine the method from multiple angles, including interesting phenomena like how popular or “welcome” certain memorized content tends to be.

**Weaknesses:**

- I’m really unsure about the writing style of this work. The method described in the main text feels more like a high-level idea,  very rough and vague. Almost all the actual details are buried in the appendix. I’m not sure if this kind of writing is acceptable, but honestly, it made it super hard for me to understand the method clearly,  so hard that I couldn’t even judge whether the approach is reliable or not. I think the appendix should only support the main text, not carry most of the technical details.

    - Specifically, the five steps in the RECAP method require very careful reading to fully grasp. Figure 2 is just a functional overview, even with the main text, it’s still confusing because of some unclear keyword usage. Maybe adding a concrete example directly into the Figure 2 flowchart would help a lot. Also, the four diagrams in Figure 2 — the left side seems to show related data, but then suddenly on the right, BERT and ELMo appear. Are those meant to represent language models (I think maybe it’s Parrot BERT?)? The logic connecting them isn’t clearly explained in the figure.

- I’m also not convinced that using ROUGE-L is the right way to measure an LLM’s ability to reproduce text. ROUGE-L looks at the longest common subsequence between two texts. it allows skipping words. But when we’re talking about copyright, we usually care about whether the model can reproduce the text *exactly*, or at least reproduce a long enough chunk to count as infringement [1]. If that’s the case, maybe we should try using Word Error Rate, like in speech recognition tasks?

    - Also, how do we interpret the ROUGE-L scores? Intuitively, a higher score means stronger evidence that the LLM remembers the book content. But since there’s a FEEDBACK AGENT involved, how can we tell whether the result comes from the model’s actual memory — or just from being guided by the agent?

- And one last thing, why doesn’t Table 1 show the DSP + Jailbreak results like Table 2 does?


[1] Copyright violations and large language models

**Questions:**

see weakness

---

> ### Author Response · Authors · 2025-11-20
>
> Dear Reviewer,
>
> Thank you very much for your valuable feedback and comments. Below, we address each of your questions. We would also like to note that we have already updated the manuscript to incorporate many of your suggestions.
>
> > *The method description in the main text feels high-level and vague; most technical details are buried in the appendix.*
> >
>
> We understand this concern. The original page limit made it challenging to include the full level of technical detail we intended in the main body. In the revised submission, we have expanded Sections 3.1–3.5 to provide clearer descriptions of each module, strengthened the explanations connecting the modules, and added key technical details that previously appeared only in the appendix. The main text should now offer a more complete understanding of the pipeline.
>
> > *The flowchart could benefit from concrete examples; the appearance of BERT/ELMo icons is confusing*
> >
>
> We thank you for this suggestion. We have improved the flowchart figure to be much clearer.
>
> Regarding the icons: BERT and ELMo were intended only as visual placeholders representing the *papers category* in our dataset (just as book covers represent the *books category*). They have no connection to the Parrot-BERT. To avoid confusion, we replaced these icons with neutral document symbols.
>
> > *ROUGE-L is not ideal for verbatim reproduction because it allows skipping words; copyright concerns require exact reproduction. How should the scores be interpreted?*
> >
>
> We agree that ROUGE-L indeed allows non-contiguous matches. However, in practice, high ROUGE-L values still require substantial overlap at the token level. The very high scores we observe in some cases (e.g., ROUGE-L ≈ 0.7) would not be possible without strong word alignment between the generated and the reference passages.
>
> To complement this, in Appendix T we provide an *exact-match style* analysis based on counting how many 40-token passages each model reproduces with up to 0 mismatches. As shown, for Claude-3.7, with RECAP we extract an average of 500 such passages from each copyrighted book **with zero tolerance**, reinforcing that the RECAP reveals genuine verbatim memorization.
>
> We chose not to make “number of verbatim passages” the primary metric because absolute counts depend strongly on book length; ROUGE-L provides a book-length–agnostic summary.
>
> > *How can we know whether high ROUGE-L comes from actual model memory vs. guidance from the Feedback Agent?*
> >
>
> We believe to have carefully evaluated this possibility, and two observations suggest that feedback does **not** introduce contamination:
>
> 1. As shown in Section 5.6, the Feedback Agent does not make non-training books (which the model could not have seen) any more extractable than baselines. If the Feedback Agent were introducing information that would “help” the model guess content, we would expect similar improvements here, but we do not observe any.
> 2. In Appendix S, we also used an independent LLM to assess whether the feedback hints could be revealing too much information. The judge consistently classifies most hints as high-level corrections, thereby reinforcing that the feedback agent is “clean”.
>
> > *Why does Table 1 (arXiv papers) not include DSP + Jailbreak like Table 2 does?*
> >
>
> Our initial motivation for including the DSP + Jailbreak baseline was to isolate the contributions of jailbreaking vs. feedback specifically in the book extraction setting, which is the primary focus of our study. We agree that including it for arXiv would make the comparison more consistent. Following your suggestion, we have updated Table 1 in the revision to include this baseline as well.
>
> ---
>
> **Conclusions**
>
> Thank you again for your constructive comments. We believe the revisions resulting from your feedback have substantially strengthened the paper. Please let us know if any additional clarification would be helpful.

---

### Author Response · Authors · 2025-12-03
**Author Summary for the Area Chair**

Dear AC,

Thank you for taking the time to evaluate our submission.

Given this year’s unusual circumstances, we provide a short summary of each reviewer’s key points and our responses to help you quickly catch up on the state of the discussion before it was suspended.

---

Below we summarize the **main concerns** raised across reviews and how we addressed them:

1. **Method clarity.**

    One reviewer found the main-text description of RECAP (our method) too high-level.

    **We clarified and expanded Sections 3.1–3.5 and improved the pipeline figure**, so the core method is now understandable without relying on the appendix.
---
2. **Benchmark scope.**

    Reviewers asked for clearer motivation behind EchoTrace (our proposed benchmark) and noted overrepresentation of popular books.

    We clarified that EchoTrace targets **long-form, document-coherent memorization**, which existing snippet-based benchmarks do not capture, and we added a **new ablation on less-popular works** (Appendix R).
---
3. **Evaluation breadth.**

    Reviewers requested more experiments on open-source models and models with transparent training data.

    In addition to DeepSeek and Qwen, we added **new results for the fully open OLMo-2-32B**, which cleanly reflects expected exposure vs. non-exposure patterns (Appendix Q).
---
4. **Metrics and jailbreak behavior.**

    There were questions about ROUGE-L being the right metric and about whether the Jailbreaker could introduce noise in the pipeline.

    We **clarified our interpretation of ROUGE-L, highlighted the existing exact-match verbatim analysis** (Appendix T), and emphasized evidence already present in the paper showing that excessive jailbreak use can slightly reduce extraction quality (Appendix U.1).

---
---
As for **positive points**, reviewers also highlighted several consistent strengths:

1. The **significance and timeliness** of the problem.
2. The **effectiveness of the RECAP method**, which delivers *“clear and statistically grounded improvements across four major model families”.*
3. The **utility of the EchoTrace benchmark**, noted as “*a valuable resource for future work*”.
4. The **depth of the experimental evaluation**, praised as “*well-designed*”, “*comprehensive*”, and “*thorough*”.

---

We hope this summary is helpful, and we thank you once again for your time and attention.

---

### Meta-Review · Area_Chair_N4Hu · 2025-12-28

**Summary:**

The reviewers generally agree this paper tackles an important and timely problem—auditing whether LLMs can reproduce copyrighted training data—and they find the experimental effort (including ablations) substantial.

My main remaining concern is what the primary evaluation is actually measuring. RECAP’s agentic loop intertwines (i) exposure/memorization with (ii) instruction-following capacity and (iii) alignment/refusal/jailbreak dynamics. As a result, cross-model comparisons are difficult to interpret as “more/less memorization”: a model that is simply better at following iterative correction hints, revising outputs over multiple rounds, or refusing less can achieve higher ROUGE-L even if underlying memorization is similar. Within a fixed target model, RECAP-vs-baseline gains are less sensitive to this issue; however, many of the paper’s motivations and takeaways implicitly invite cross-model “memorization” interpretations that are not cleanly supported without disentangling experiments.

The rebuttal and revisions improve presentation clarity and add helpful analyses (including stronger controls/open or transparent-model evidence and additional verbatim-style reporting), which partially strengthen the case that the feedback loop is not trivially contaminating outputs. However, these additions do not resolve the central interpretability confound between memorization and controllability. Combined with the continued reliance on closed-source models where training inclusion cannot be verified, I do not think the current evidence cleanly supports the paper’s implied conclusions and motivations. More experiments are needed.

Due to unclear evaluation meaning for the headline claims, it is hard to recommend acceptance. I encourage resubmission with capability-controlled disentangling experiments and clearer positioning of motivations.

**Reviewer Concerns:**

Concerns addressed

Clarity: The rebuttal claims substantial expansion of the core method sections and a clearer pipeline figure

Prompt-length confound issue: The authors point to a token-budget ablation and explain how prompt length can be controlled via the summary module

Feedback agent contamination: new analysis partially mitigate the worry that the feedback loop itself leaks the reference

Partially remaining concerns

Jailbreak robustness: The rebuttal argues a static jailbreak is near-ceiling today and presents a stress-test suggesting overuse can degrade extraction quality. However, robustness of this module is still only partially characterized.

Metric choice: The added verbatim-style reporting strengthens the story, but ROUGE-L remains a somewhat indirect proxy for “copyright-relevant” verbatim reproduction and can still blur interpretability (especially across models).

Closed-source evaluations: The addition of results on transparent/open models meaningfully helps validate the procedure, but for closed-source models the paper still cannot verify that specific copyrighted sources were in training. This remains a core limitation for any strong claims about membership/exposure on those systems.

Concerns outstanding:

What RECAP is actually measuring: see metareview

**Reviewer Scores:**

Ytx5/os15 may maintain or lift scores
p8p2/pcch may maintain scores

---

### Decision · Program_Chairs · 2026-01-26

Reject